# WBP2 inhibits microRNA biogenesis via interaction with the microprocessor complex

Hossein Tabatabaeian[1,2], Shen Kiat Lim[1,3], Tinghine Chu[1,4], Sock Hong Seah[5], Yoon Pin Lim[1,4,6]

**WBP2 is an emerging oncoprotein with diverse functions in breast tumorigenesis via regulating Wnt, epidermal growth factor receptor, estrogen receptor, and Hippo. Recently, evidence shows that WBP2 is tightly regulated by the components of the miRNA biogenesis machinery such as DGCR8 and Dicer via producing both WBP2's 3′UTR and coding DNA sequence-targeting miRNAs. This led us to hypothesize that WBP2 could provide a feedback loop to the biogenesis of its key upstream regulators by regulating the microprocessor complex activity. Indeed, WBP2 suppressed microprocessor activity by blocking the processing of pri-miRNAs to pre-miRNAs. WBP2 negatively regulated the assembly of the microprocessor complex via physical interactions with its components. Meta-analyses suggest that microprocessor complex components, in particular DGCR8, DDX5, and DEAD-Box Helicase17 (DDX17), have tumor-suppressive properties. 2D and 3D in vitro proliferation assays revealed that WBP2 blocked the tumor-suppressive properties of DGCR8, a key component of the microprocessor complex. In conclusion, WBP2 is a novel regulator of miRNA biogenesis that is a known dysregulated pathway in breast tumorigenesis. The reregulation of miRNA biogenesis machinery via targeting WBP2 protein may have implications in breast cancer therapy.**

## Introduction

Breast cancer remains the most prevalent and the leading cause of cancer-related deaths in women worldwide (1). There are about 2.1 million newly diagnosed breast cancer cases in females annually, which accounts for roughly a quarter of cancer cases among women (1). Despite remarkable improvements in our understanding and management of breast cancer, there is a pertinent need for a better understanding of the molecular etiology of breast cancer.

WW domain-binding protein 2 (WBP2) is an emerging oncoprotein profoundly implicated in a variety of transduction systems,

such as Wnt, Hippo, epidermal growth factor receptor (EGFR) and steroid signaling pathways, and human cancers including breast malignancies. Clinically, high WBP2 expression has been observed in >85% of the breast tumor tissues as compared with normal (2). The expression of WBP2 correlates positively and significantly with tumor size and grade, and negatively with disease-free and overall survival of breast cancer patients including those with HER2-positive breast cancer (2, 3). These data highlight the importance of WBP2 to early development of breast cancer and its aggression.

WBP2, a transcription coactivator initially identified as the cognate ligand of yes-associated protein (YAP) protein (4), was discovered to be associated with breast cancer progression in 2007 (5). This protein exerts its oncogenic properties via diverse modes of action through interacting with WW-containing and non-WW-containing proteins in Wnt (6, 7), estrogen receptor (ER)/progesterone receptor (8, 9), PI$_3$K/Akt (10, 11), EGFR (6), and Hippo (12) signaling pathways culminating in phenotypes associated with cell growth, proliferation, anchorage-independent growth, invasion, and migration (6, 13).

WBP2 is tightly regulated at multiple levels. Upstream stimulatory factor 1 (USF-1) protein is the sole transcription factor identified to date that positively regulates the transcription of the *WBP2* gene in response to insulin stimulation (14). WBP2 protein expression is further regulated posttranslationally by ITCH E3-ubiquitin ligase, which down-regulates WBP2 expression; and by tyrosine phosphorylation (2) that promotes its cytoplasmic to nuclear translocation in response to estrogen and Wnt ligands via EGFR crosstalk (6). A number of miRNAs have also been identified to regulate WBP2 at the post-transcriptional level. For example, the 3′UTR region of WBP2 has been demonstrated to be targeted by miR-206, miR-613, and miR-23a in breast cancer (11, 15, 16), and miR-458 in hepatocellular carcinoma (17).

miRNAs are small noncoding RNAs known to be critical regulators of gene expression and cell fate (18). While miRNAs can have either oncogenic or tumor-suppressive roles in cancer, a prevailing observation is the global suppression of miRNAs levels in human cancers (19, 20, 21). miRNAs are first transcribed as primary miRNAs (Pri-miRNAs). These immature sequences are then trimmed by the

[1]Department of Biochemistry, Yong Loo Lin School of Medicine, National University of Singapore, Singapore    [2]Cancer Science Institute of Singapore, National University of Singapore, Singapore    [3]School of Biological Sciences, Nanyang Technological University, Singapore    [4]National University of Singapore Graduate School for Integrative Sciences and Engineering, National University of Singapore, Singapore    [5]Mechanobiology Institute, National University of Singapore, Singapore    [6]National University Cancer Institute, Singapore

Correspondence: bchlyp@nus.edu.sg

 

microprocessor complex in the nucleus, to generate the precursor miRNAs (Pre-miRNAs). The pre-miRNAs are further processed in the cytoplasm by Dicer complex to produce mature miRNAs. In tight interaction with RNA-induced silencing (RISC) complex, mature miRNAs regulate gene expression via either translational repression or mRNA degradation (22).

The microprocessor complex constitutes of Drosha, DiGeorge Critical Region 8 (DGCR8), DEAD-box helicase 5 (DDX5), and DEAD-Box Helicase17 (DDX17) proteins (23). In addition to these core components, a number of auxiliary molecules have been recently reported to be important in the processing of Pri-miRNAs into Pre-miRNAs, for example, BRCA1 (23), Smads (24), Myc (25), and p53 (26). Recently, our laboratory showed that WBP2 is regulated by both 3′UTR and coding DNA sequence-targeting miRNAs and that Dicer or DGCR8 modulates WBP2 expression under the influence of mammalian sterile 20-like (MST)/Hippo signaling (16). Taken together, we hypothesize that WBP2 regulates miRNA biogenesis via a negative feedback loop.

Here, we show that WBP2 negatively regulates the microprocessor complex activity through interacting with the microprocessor complex components.

## Results

### WBP2 inhibits pri-miRNA processing by regulating the microprocessor complex

To test the hypothesis that WBP2 negatively regulates miRNA biogenesis, we first examined the potential role of WBP2 in regulating the microprocessor complex activity. To this end, a microprocessor activity reporter construct, containing pri-miR-125-b1 or pri-miR-205 cloned downstream of Renilla luciferase in psiCHECK2 plasmid, was used. These primary sequences were selected as the feasibility of the constructs was previously validated (27). Any changes in the microprocessor complex activity affect the stability of the transcribed sequence, and thereby the luciferase activity (Fig S1). The Firefly/Renilla ratio was used as a measure of the microprocessor complex activity upon WBP2 overexpression or siRNA knockdown in MCF-7 cells. This cell line was used because of the moderate expression of WBP2.

As shown in Fig 1Ai and ii, WBP2 depletion by specific siRNA resulted in the significant elevation of microprocessor complex activity. Consistently, WBP2 overexpression decreased microprocessor complex activity. DGCR8, which is a key component of the microprocessor complex, was depleted and used as a positive control. Although DGCR8 knockdown showed a consistent decrease in the microprocessor complex activity, the efficiency of DGCR8 siRNA was not robust. Thus, we used DGCR8 overexpression in the subsequent assays, which significantly and consistently increased the microprocessor activity. Expectedly, the negative control psiCHECK2 plasmid failed to alter the microprocessor complex activity (Fig 1B). These findings suggest that WBP2 is a negative regulator of the microprocessor complex.

To ensure that the finding could be recapitulated in other breast cancer cell lines, WBP2 was overexpressed in BT-474 cells that endogenously lack the WBP2 protein expression, as well as over-expressed and silenced in T47D cells that have a moderate WBP2 expression level and the microprocessor complex assay performed. As can be seen in Fig 1C and D, WBP2 consistently and negatively regulated the putative microprocessor complex activity in BT-474 and T47D cell lines. Taken together, WBP2 potentially regulates microprocessor complex activity in a negative fashion in both breast cancer cells.

Despite the above data, it is not known if WBP2 destabilizes the Renilla–pri-miRNA transcript via manipulating the activity of the microprocessor complex or any unknown alternative way. To clarify this, the effect of WBP2 on the key components of the microprocessor complex was studied. As shown in Fig 2A, WBP2 decreased the DGCR8-driven microprocessor complex activity in MCF-7 cells. Moreover, while Drosha knockdown significantly decreased the microprocessor complex activity, co-silencing Drosha with WBP2 resulted in a partial recovery of the complex activity (Fig 2B). Likewise, WBP2 diminished the DDX5-driven microprocessor complex activity (Fig 2C). These data collectively indicate that WBP2 potentially inhibits miRNAs biogenesis by suppressing the function of the microprocessor complex.

To authenticate the finding, we examined the inhibitory role of WBP2 on microprocessor complex activity with another method. Any regulation of the microprocessor complex activity should affect the production of pre-miRNA and therefore the pre-/pri-miRNA ratio. In this assay, the total RNA was first extracted. This was followed by the subpopulation isolation of different RNA species, that is, >200-nucleotide and <200-nucleotide extracts (Fig S2A).

As representatives, four miRNAs, namely, miR-19a, miR-19b, and miR-23a and miR-205 were used to investigate the pre-/pri-miRNA expression level ratio. miR-19a and miR-19b were previously determined to be incapable of targeting WBP2, whereas miR-23a had been proven to target WBP2 (16), miR-205 was randomly selected. Fig 2D illustrates that WBP2 knockdown increased the ratio, whereas WBP2 overexpression decreased the ratio for all the miRNAs studied in MCF-7 cells. DGCR8 overexpression was used as the positive control, which resulted in the expected elevation of pre-/pri-miRNA signal ratio. This verifies that the assay has been robustly performed. Interestingly, the effect of WBP2 depletion and overexpression resulted in the increased and decreased expression of the mature transcripts of the selected miRNAs, respectively (Fig 2E). The finding confirmed that WBP2 negatively regulates the levels of pre-miRNA, leading to the diminished expression of mature miRNAs. The quality controls (QCs) for WBP2 knockdown and WBP2/DGCR8 overexpression are shown in Fig S2B.

Taken together, the findings generated by two independent techniques, that is microprocessor complex assay and qPCR-based pre-/pri-miRNA ratio, support the notion that WBP2 protein negatively regulates pri-miRNA processing into pre-miRNA by inhibiting the microprocessor complex activity.

### Nuclear localization of WBP2 enhances its negative effect on the microprocessor complex

Because miRNA bioprocessing takes place in the nucleus, we hypothesize that disruption of the nuclear localization of WBP2 would impede its effects on microprocessor complex activity. To this end,

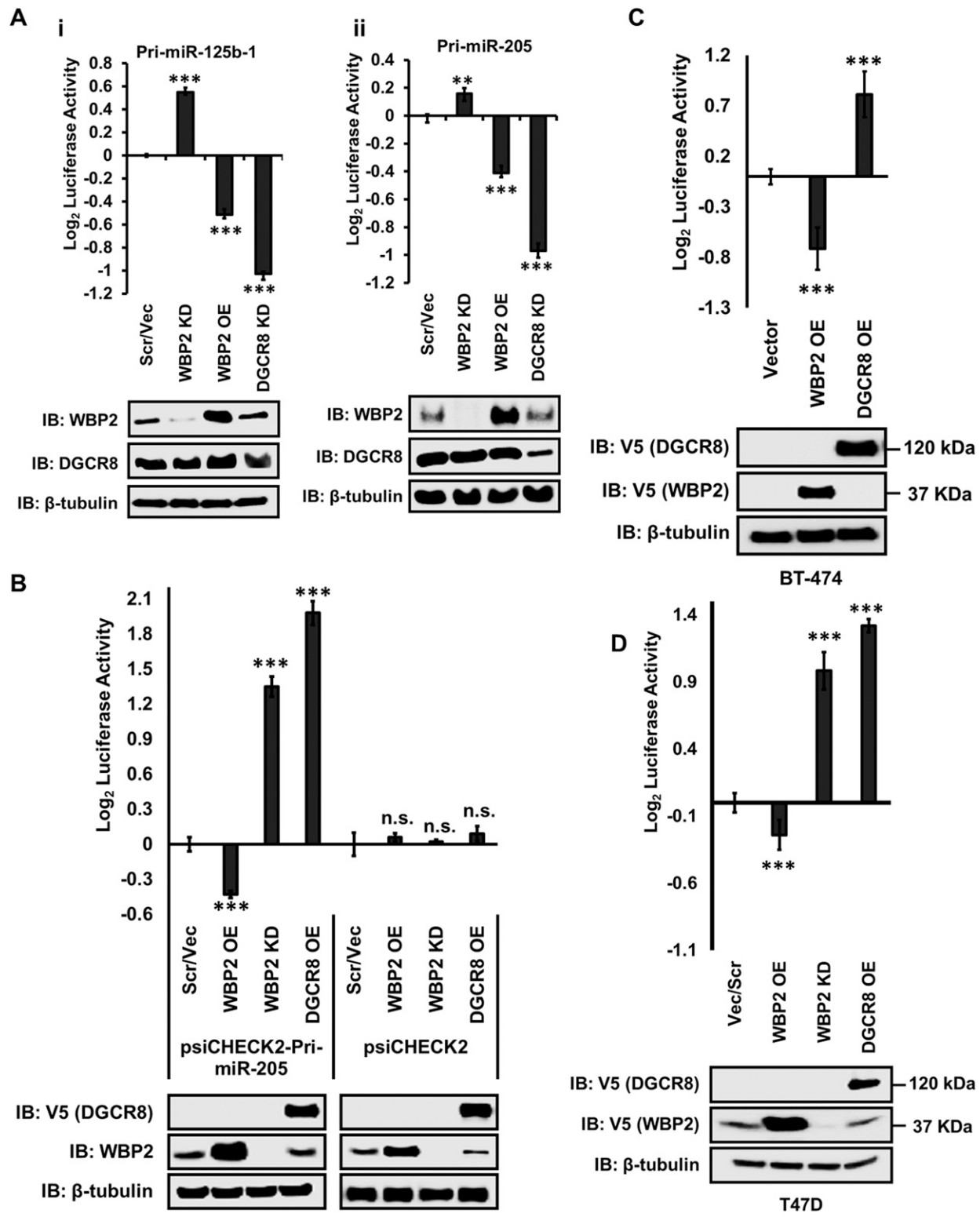

**Figure 1. WBP2 negatively regulates the activity of the microprocessor complex.**
**(A)** Analysis of microprocessor complex activity by pri-miR-125b-1 (i) and pri-miR-205 (ii) constructs-based microprocessor complex assay in MCF-7 cells. DGCR8 depletion was used as the positive control. **(B)** WBP2 and DGCR8 negatively and positively regulate the processing of pri-miR-205 construct, while they do not regulate the empty vector in MCF-7 cells. DGRC8 overexpression was used as the positive control. **(C, D)** Similar to MCF-7 cells, WBP2 negatively regulates microprocessor complex activity in BT-474 and (D) in T47D cells using pri-miR-205 construct-based microprocessor complex activity assay. DGRC8 overexpression was used as the positive control.

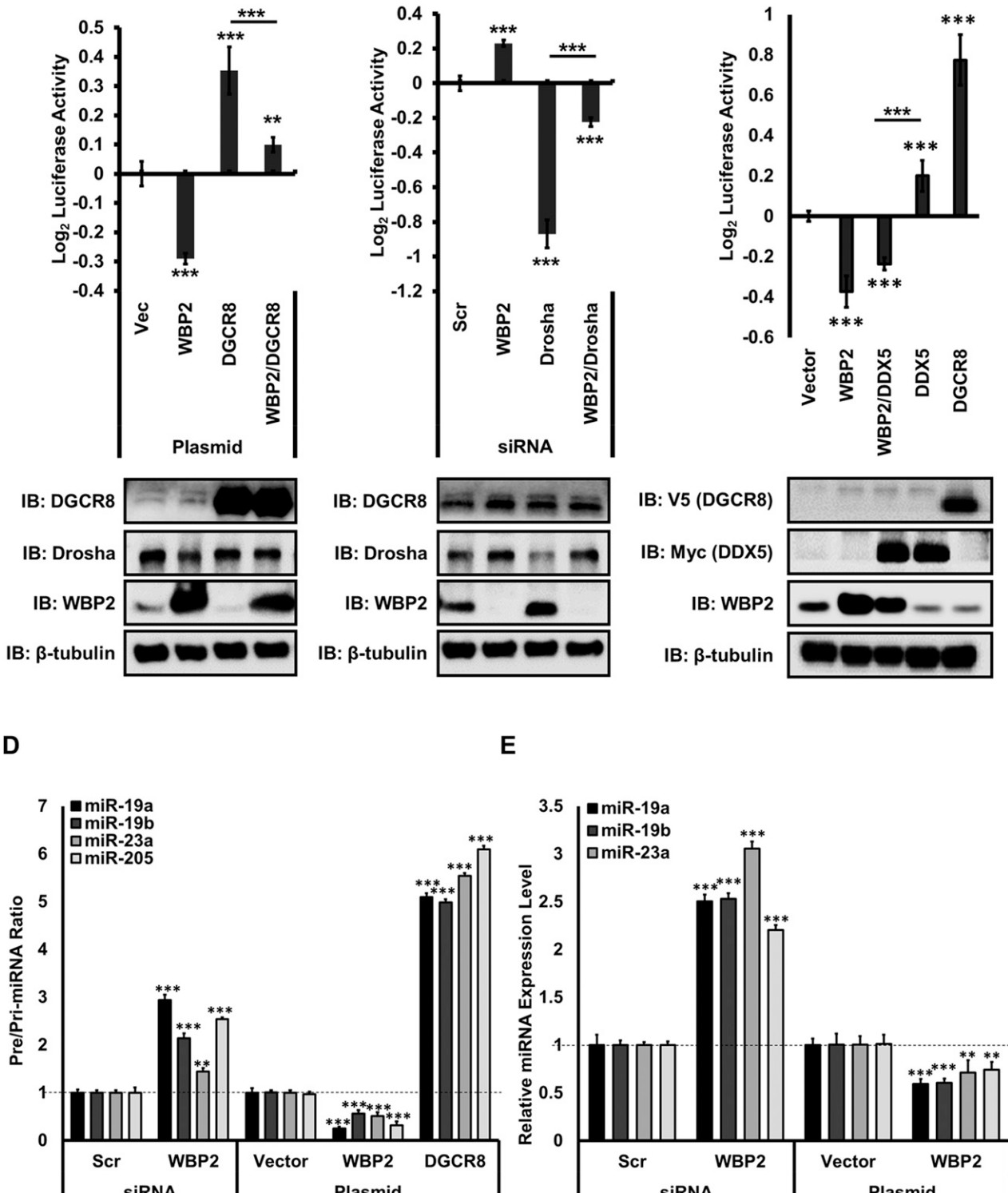

**Figure 2. WBP2 regulates the processing of pri-miRNA specifically via the microprocessor complex.**
**(A)** WBP2 attenuates the DGCR8-driven microprocessor complex activity. **(B)** WBP2 silencing partially recovers the suppressed activity of the microprocessor complex, caused by Drosha silencing. **(C)** WBP2 attenuates DDX5-driven microprocessor complex activity in MCF-7 cells. pri-miR-205 construct–based microprocessor complex activity assay was used. **(D)** Analysis of microprocessor complex activity based on the pre-/pri-miRNA ratio upon WBP2 silencing and overexpression using qPCR technique in MCF-7 cells. **(E)** Analysis of mature miRNA expression upon WBP2 silencing and overexpression using qPCR technique in MCF-7 cells.

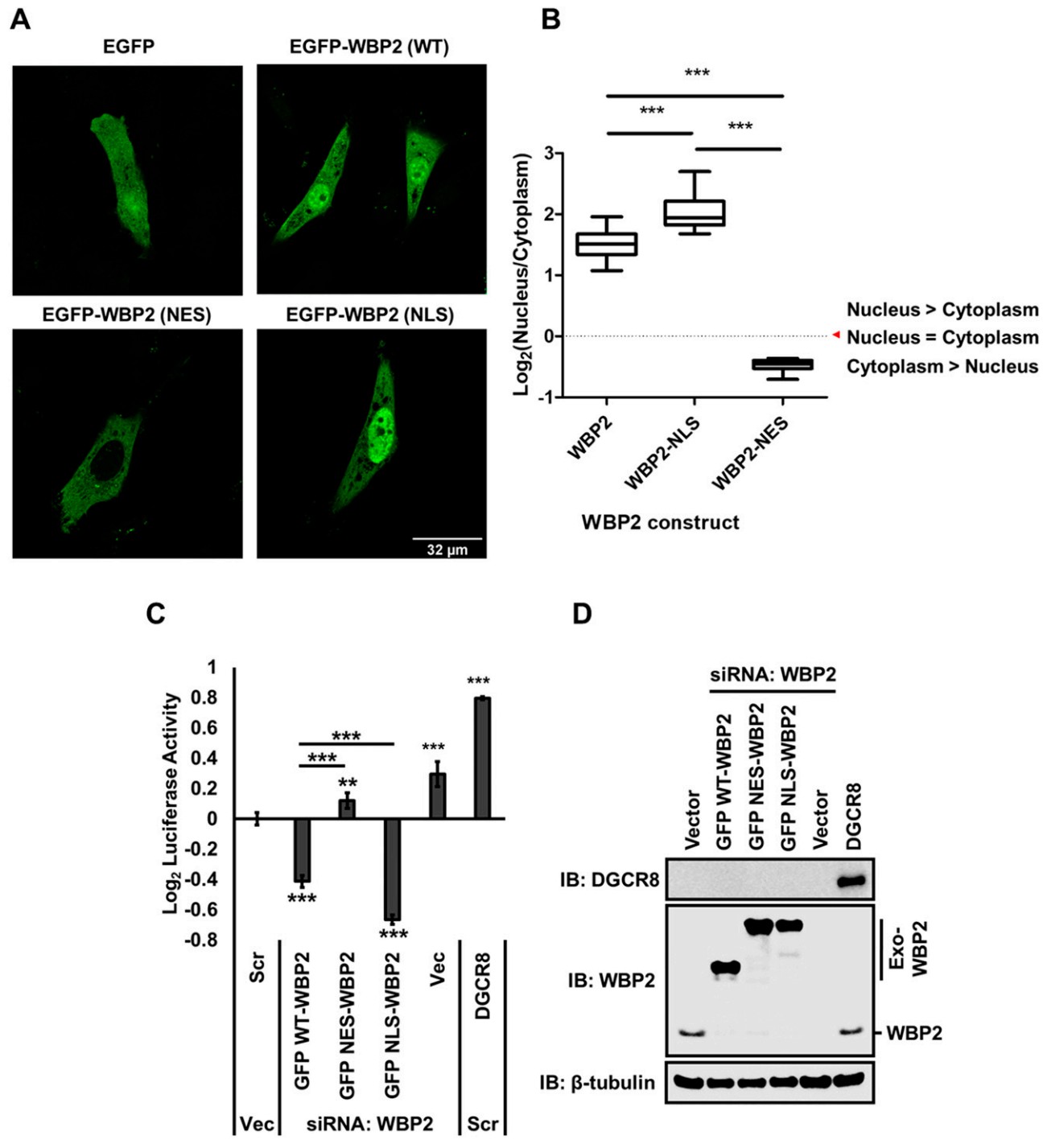

**Figure 3. Nuclear localization of WBP2 modifies its capability in regulating the microprocessor complex.**
**(A)** Analysis of nuclear export signal (NES)- and NLS-tagged WBP2 localization in MCF-7 cells using immunofluorescence method (n = 10 cells). **(B)** NLS-WBP2 localizes significantly in the cytoplasm, whereas NES-WBP2 localizes in the nucleus. **(C)** NES-GFP-WBP2 is unable to inhibit the microprocessor complex activity when the endogenous WBP2 is depleted using pri-miR-205 construct–based microprocessor complex assay in MCF-7 cells. Whereas NLS-GFP-WBP2 significantly inhibits the microprocessor complex greater than wild-type-GFP-WBP2. **(D)** The wild-type/NES/NLS-GFP–tagged exogenous WBP2 are overexpressed properly, whereas the endogenous WBP2 is silenced. DGCR8 is overexpressed as a positive control of the microprocessor complex activity assay.

the NLS or nuclear export signal (NES)–tagged WBP2 constructs were transfected in MCF-7 cells. Immunofluorescence method was used to track the localization of WBP2 constructs (Fig 3A). Quantitative analysis showed that NES-tagged WBP2 is mostly localized

in the cytoplasm, whereas the NLS-tagged WBP2 is in the nucleus predominantly (Fig 3B). The endogenous WBP2 was first depleted using siRNAs to focus on the exogenous NES/NLS-tagged WBP2. As shown in Fig 3C, the wild-type GFP-WBP2—as expected—reduced

the microprocessor complex activity, whereas the NES-GFP-WBP2 was unable to diminish. In contrast, NLS-GFP-WBP2 inhibited the microprocessor activity significantly more than the wild type. QC in Fig 3D shows that both endogenous WBP2 silencing and overexpression of different WBP2 constructs were successful in MCF-7 cells. These findings support the role of WBP2 as a negative regulator of miRNA biogenesis via its action on the microprocessor complex in the nucleus. These data can be further supported by the co-immunofluorescence method.

**WBP2 physically interacts with microprocessor complex**

Given the role of WBP2 in transcription regulation, we tested the hypothesis that WBP2 negatively regulates microprocessor complex by examining if it suppresses the expression of microprocessor complex components. Immunoblotting reveals that the protein level of Drosha, DGCR8, DDX5, and DDX17 did not change significantly upon overexpression (Fig 4Ai) or silencing (Fig 4Aii) of WBP2 in MCF-7 and T47D cells.

The observation that WBP2's nuclear localization is required for its action on the microprocessor complex suggests that physical interaction is a possible mode of action. Thus, co-IP was used to assess this hypothesis. As the nuclear expression of WBP2 was too low in both cell lines, the WBP2 and microprocessor complex components were co-overexpressed to capture the interactions effectively (Fig S3). As shown in Fig 4Bi and ii, WBP2 pull-down in the nuclear subfraction revealed its co-existence with all the microprocessor complex components in T47D cells, whereas the interaction was detected with Drosha, DGCR8, and DDX5 in MCF-7 cells. Reciprocally and consistently, immunoprecipitation of microprocessor complex components in the nuclear subfraction showed a robust interaction of all microprocessor complex components with WBP2 in T47D, whereas only DGCR8 and DDX5 showed interaction with WBP2 in MCF-7 cells (Fig 4C and D). The reason for observing different complexes in different cell lines could be due to the much lower nuclear expression of WBP2 in MCF-7 cells as compared with T47D cells, which technically affected the capture of WBP2's interactions inversely. The difference might also suggest a diversity of interactions within the microprocessor complex due to cellular heterogeneity. As can be seen in Fig 4D, neither DDX5 nor DDX17 pull-down was able to capture Drosha in MCF-7 cells. The same came true when we pull-down Drosha and probed for DDX5 and DDX17. However, the meaningful interactions between Drosha, DDX5, and DDX17 were detected in T47D cells (Fig 4C). These imply that, regardless of WBP2's role, the incorporation of DDX5 and DDX17 to the microprocessor complex is less notable in MCF-7 cells. Collectively, the co-IP results demonstrated the interaction of WBP2 with microprocessor complex components. Because DGCR8 and DDX5 were the common interacting partners of WBP2 in both cell lines, we focused on these proteins in subsequent studies.

**WBP2 suppresses the assembly of the microprocessor complex**

We showed a tight interaction of WBP2 with the microprocessor complex. However, it was not known whether this interaction is responsible for the WBP2 inhibitory effect on this complex. To answer this question, we checked the intra-microprocessor complex

interactions among its different key components upon WBP2 overexpression and knockdown in T47D (Fig 5Ai and Bi) and MCF-7 (Fig 5Ci and Di) cells. Interestingly, DGCR8 immunoprecipitation showed decreased interaction with Drosha, DDX5, and DDX17 in the nucleus subfraction upon WBP2 overexpression (Fig 5Aii). Consistently, the Drosha pull-down revealed diminished interaction with DGCR8, DDX5, and DDX17 (Fig 5Aiii) upon WBP2 overexpression in T47D cells. Similar results were observed upon WBP2 overexpression in MCF-7 cells (Fig 5Bii). Consistently, DGCR8 immunoprecipitation demonstrated higher microprocessor complex assembly upon WBP2 knockdown via elevating the interaction between DGCR8, Drosha, and DDX5 in T47D (Fig 5Cii and iii) and MCF-7 (Fig 5Dii) cells. In nutshell, WBP2 could inhibit the microprocessor complex activity via interacting with its component(s), leading to suppression of microprocessor complex assembly.

It is known that WBP2 mainly interacts with its partners via its PPxY (PY) motif (4). Hence, we tested PPxY-mutant WBP2 constructs to determine which motif might be responsible for the binding. Performing V5-WBP2 pull-down upon overexpression of different combinations—PY1 Mut, PY2 Mut, PY3 Mut, PY1/2 Mut, PY1/3 Mut, PY2/3 Mut, and PY1/2/3 Mut—showed that PPxY motif was not involved in WBP2/DGCR8 interaction (Fig S4A–C), suggesting that WBP2 negatively regulates the microprocessor complex activity independent of its PY motifs. Interestingly, the WBP2 PY mutants—alone or in combination—did similarly reduce the luciferase activity. Other WBP2 domains may be responsible for such interaction; however, it remained to be tested because of technical limitations.

**WBP2 abolished the tumor-suppressive effects of the microprocessor complex**

The net tumor-suppressive role of miRNAs in human cancers has been reported (19, 20, 21). To attain higher resolution insight into this, we analyzed The Cancer Genome Atlas (TCGA) database to examine the expression pattern of the microprocessor complex components in breast cancer. In line with previous findings, down-regulation of DDX5, DDX17, and DGCR8 transcript levels in breast cancer was observed (Fig 6A–C). Drosha did not show a similar trend (Fig S5). The reason is unclear, but it may be due to the observation that *Drosha* is regulated mainly at the posttranslational and not RNA level (28, 29).

Next, we examined how the expression levels of *DDX5*, *DDX17*, and *DGCR8* genes correlate with breast cancer patients' survival. The results show that *DGCR8*, *DDX5*, and *DDX17* individually correlated with higher relapse-free survival in breast cancer, hazard ratio (HR) = 0.78 (95% CI: 0.7–0.87, logrank $P$-value = 7.7 × 10$^6$), HR = 0.72 (95% CI: 0.62–0.84, logrank $P$-value = 3.3 × 10$^5$), HR = 0.66 (95% CI: 0.59–0.74, logrank $P$-value = 2.1 × 10$^{13}$), respectively (Fig 6D–F). Combined *DDX5*/*DDX17*/*DGCR8* expression profiles as a signature showed a stronger effect, HR = 0.47 (95% CI: 0.4-0.55, logrank $P$-value = 1 × 10$^{16}$) (Fig 6G), meaning that lower expression of this set of signature genes correlated with ~2 times increase in the survival of breast cancer patients.

Given the tumor-suppressive role of the microprocessor complex components and the putative regulatory role of WBP2 in microprocessor complex function, we performed 2D and 3D growth

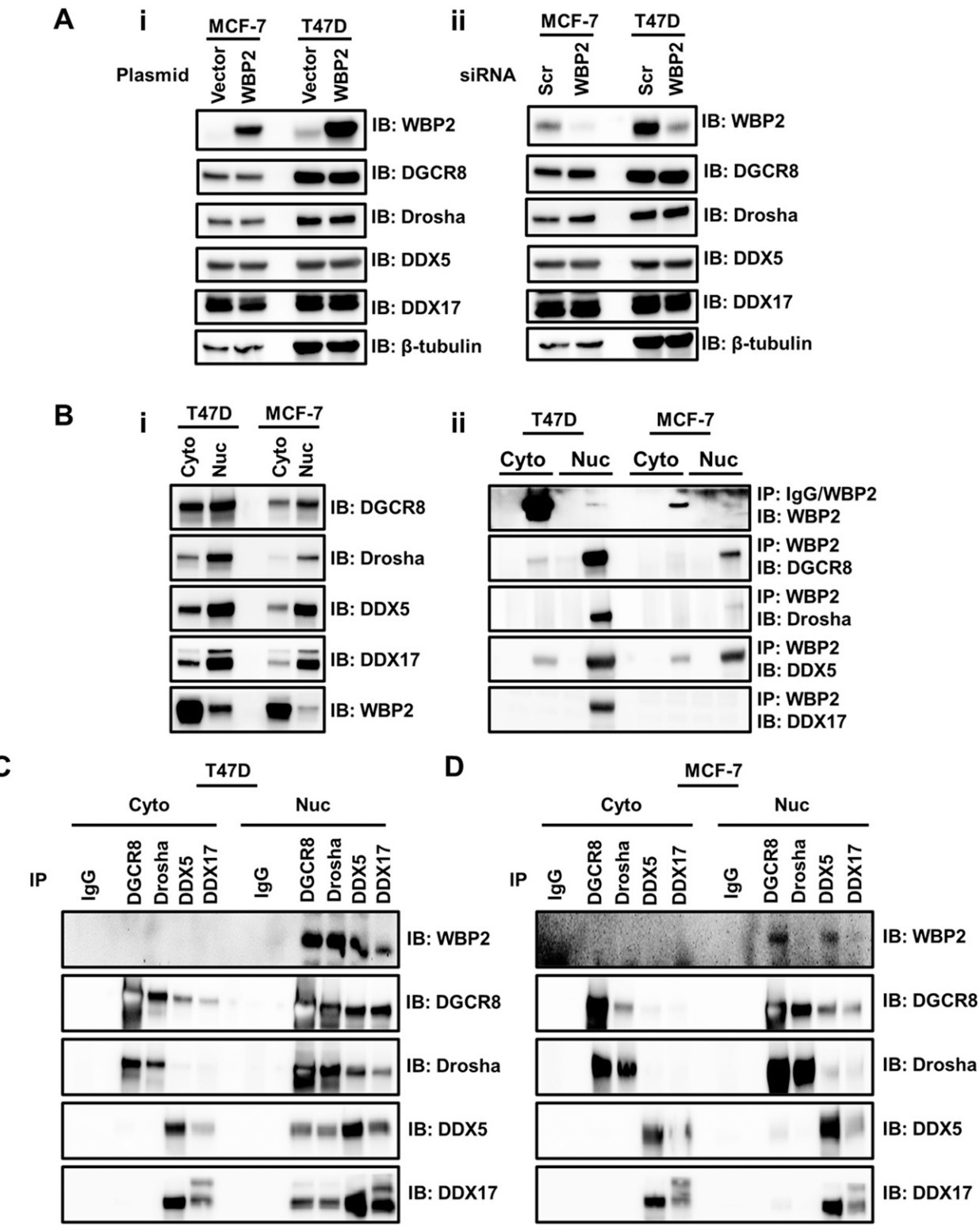

**Figure 4. WBP2 does not regulate the expression level but interacts with the microprocessor complex components.**
**(A)** WBP2 overexpression (i) or depletion (ii) do not affect the expression level of DGCR8, Drosha, DDX5, and DDX17 in MCF-7 and T47D cells. **(B-i)** Analysis of WBP2, DGCR8, DDX5, and DDX17 expression levels in the cytoplasm and nucleus of MCF-7 and T47D cells. **(B-ii)** Analysis of WBP2's interaction with the microprocessor complex components in subcellular fractionations of MCF-7 and T47D cells, upon WBP2 pull-down. **(C)** Reciprocal analysis of WBP2's interaction with the microprocessor complex components in subcellular fractions of MCF-7 cells, upon DGCR8, Drosha, DDX5, or DDX17 pull-down. **(D)** Reciprocal analysis of WBP2's interaction with the microprocessor complex components in subcellular fractions of T47D cells, upon DGCR8, Drosha, DDX5, or DDX17 pull-down.

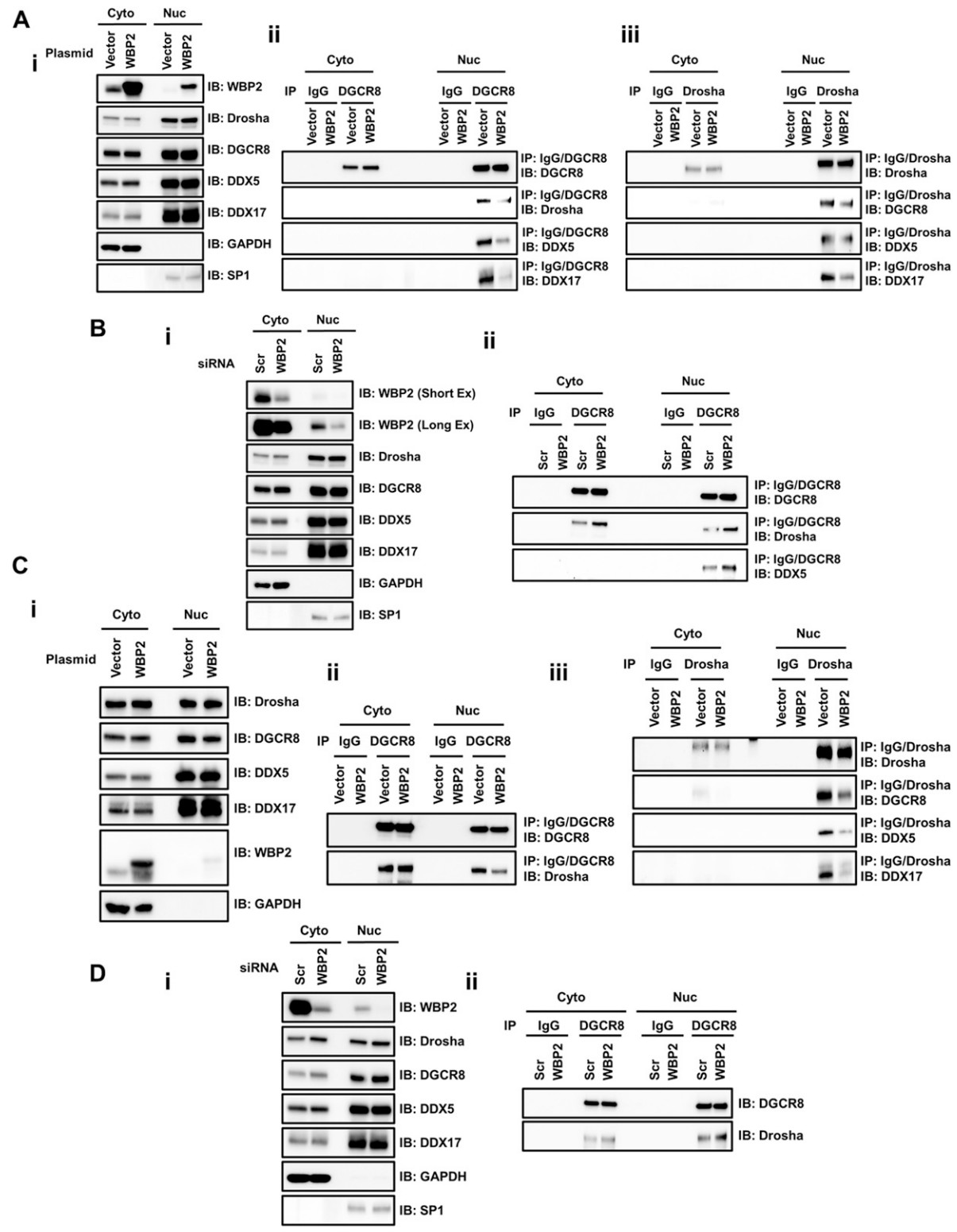

**Figure 5. WBP2 negatively regulates the assembly of the microprocessor complex.**
**(A-i)** QC for the overexpression of WBP2 in different subcellular fractions and the lack of effect on the expression level of the microprocessor complex components in T47D cells. **(A-ii)** Analysis of the microprocessor complex assembly upon WBP2 overexpression upon DGCR8 pull-down in T47D cells. **(A-iii)** Analysis of the microprocessor complex assembly upon WBP2 overexpression upon Drosha pull-down in T47D cells. **(B-i)** QC for the depletion of WBP2 in different subcellular fractions and the lack of effect on the expression level of the microprocessor complex components in T47D cells. **(B-ii)** Analysis of the microprocessor complex assembly upon WBP2 knockdown upon DGCR8 pull-down in T47D cells. **(C-i)** QC for the overexpression of WBP2 in different subcellular fractions and the lack of effect on the expression level of the

assays to investigate whether WBP2 proto-oncogene interferes with the microprocessor complex function in breast cancer cell lines. As shown in Fig 6Hi, DGCR8 depletion resulted in increased 3D proliferation rates in MCF-7 cells, whereas silencing the WBP2 expression slightly decreased it. Interestingly, WBP2 and DGCR8 double knockdown suppressed the spheroid growth in these cells, suggesting that WBP2 could invert the DGCR8's function potentially via manipulating the miRNA processing. Consistently, overexpression of DGCR8 and WBP2 led to the suppressed and promoted MCF-7 cell growth, respectively; and more importantly, co-overexpression of WBP2 and DGCR8 rescued the DGCR8-driven growth inhibition in these cells (Fig 6Hii). The same functional observations were observed in T47D cells using 3D proliferation assay, as shown in Fig 6Ii and ii. Of note, these data were reproducible using 2D in vitro proliferation assay in MCF-7 and T47D cells (Fig S6A and B). The QC of WBP2/DGCR8 depletion and overexpression is shown in Fig S6C and D. Together with the clinical bioinformatics analyses, these findings indicated that WBP2 could impose its oncogenic functions by inhibiting the tumor-suppressive properties of the microprocessor complex in breast cancer.

## Discussion

Besides regulation at the posttranslational and transcriptional levels, WBP2 transcripts are also tightly regulated posttranscriptionally by miRNAs, such as miR-206 (15), miR-613 (11), and miR-23a (16) in breast cancer, and miR-485 in hepatocellular carcinoma (17). Beyond these WBP2 3′UTR-targeting miRNAs, we recently showed that WBP2 is negatively regulated by coding DNA sequence-targeting miRNAs as well. Of note, we demonstrated that Dicer depletion increases the WBP2 expression drastically, indicating that WBP2 is regulated by a broad range of miRNAs (16). However, it remains unclear if WBP2 regulates the miRNA biogenesis machinery as a potential feedback loop.

Here, we report a novel function of WBP2 in negatively regulating the activity of the microprocessor complex (Fig 7) in addition to its better-known function as a transcription coactivator. This discovery was made using two different methods including microprocessor complex activity assay and qPCR-based pre-/pri-miRNA ratio assessment. The Northern blotting method could be used further to strengthen the notion that WBP2 negatively regulates the microprocessor complex activity, by examining the relative levels of selected pri-miRNA and pre-miRNA bands shown in Fig 2E.

Although WBP2 did not regulate the expression level of the microprocessor complex components, that is, DGCR8, Drosha, DDX5, and DDX17, it suppressed the activity of pri-miRNA–processing machinery via physical interactions with these components in breast cancer. Specifically, WBP2 was physically in a complex with all the components in T47D cells, whereas its interactions were only observed with DDX5 and DGCR8 in MCF-7 cells. It is unclear why, but

the cell-type-specific interaction networks may highlight the heterogeneity in breast cancer biology. Studying the WBP2/microprocessor complex in other breast cancer cell lines such as BT-474 could enlighten the constant interacting partner of WBP2 in the complex. In addition, immunoprecipitation of endogenous WBP2 followed by mass spectrometry could enhance the sensitivity of identification of WBP2's partners in the microprocessor complex and map the interaction domain/motif between WBP2 and its partners. This could rectify the potential non-specific interactions that might be detected upon the pull-down of the "exogenous" WBP2 or microprocessor complex components.

Previous reports depicted various mechanisms by which WBP2 exerts its oncogenic properties in breast cancer. WBP2 interacts with YAP/Tafazzin (TAZ) oncogenic coactivators to regulate the downstream tumorigenic pathways in the Hippo pathway (12) and with β-catenin/TCF in the Wnt pathway (2). Moreover, WBP2 promotes EGFR and $PI_3K$/Akt pathways, leading to elevation of proliferation and migration (2, 6, 11).

In this study, yet another novel oncogenic mode of action of WBP2 was discovered—an inhibitory property on the microprocessor complex activity. Despite Peric et al reported that microprocessor complex inhibition—via Drosha knockdown—induces the growth arrest (30), we showed that suppression of microprocessor complex activity via DGCR8 depletion increased both 2D and 3D in vitro cell proliferation in MCF-7 and T47D cells. This discrepancy might be due to the technical differences used in the studies; for example, different microprocessor complex components were targeted. Given such discrepancy, we sought to attain better clarity by analyzing the TCGA RNA-seq data. The analyses revealed that the key components of microprocessor complex such as DGCR8, DDX5, and DDX17 are down-regulated in breast cancer as compared with the healthy samples. However, the meta-analyses showed that Drosha was up-regulated in breast cancer. This could reconcile the discrepancy observed between the tumor-suppressive nature of DGCR8 and the oncogenic property of Drosha in breast cancer, suggesting that Drosha might be involved in other cellular complexes and functions as well. Gregory et al demonstrated that Drosha is physically involved in two different multi-protein complexes. In principle, Drosha forms a smaller complex in interaction with DGCR8 leading to the accurate biogenesis of pri-miRNAs. However, Drosha is also involved in a larger complex composing of various classes of RNA-associated proteins such as Ewing's sarcoma family of proteins (31). Although the miRNA-processing property of Drosha is not confined to the smaller complex, other functions of Drosha in the larger complex have not been conclusively elucidated. These reports collectively reflect the heterogeneous functions of Drosha in the cell, thereby the Drosha silencing-mediated growth arrest reported in Petric et al study may not necessarily show the oncogenic property of the microprocessor complex. Together with the meta-analyses data, our 2D and 3D in vitro assays support the overall tumor-suppressive function of

---

microprocessor complex components in MCF-7 cells. **(C-ii)** Analysis of the microprocessor complex assembly upon WBP2 overexpression upon DGCR8 pull-down in MCF-7 cells. **(C-iii)** Analysis of the microprocessor complex assembly upon WBP2 overexpression upon Drosha pull-down in MCF-7 cells. **(D-i)** QC for the depletion of WBP2 in different subcellular fractions and the lack of effect on the expression level of the microprocessor complex components in MCF-7 cells. **(D-ii)** Analysis of the microprocessor complex assembly upon WBP2 knockdown upon DGCR8 pull-down in MCF-7 cells.

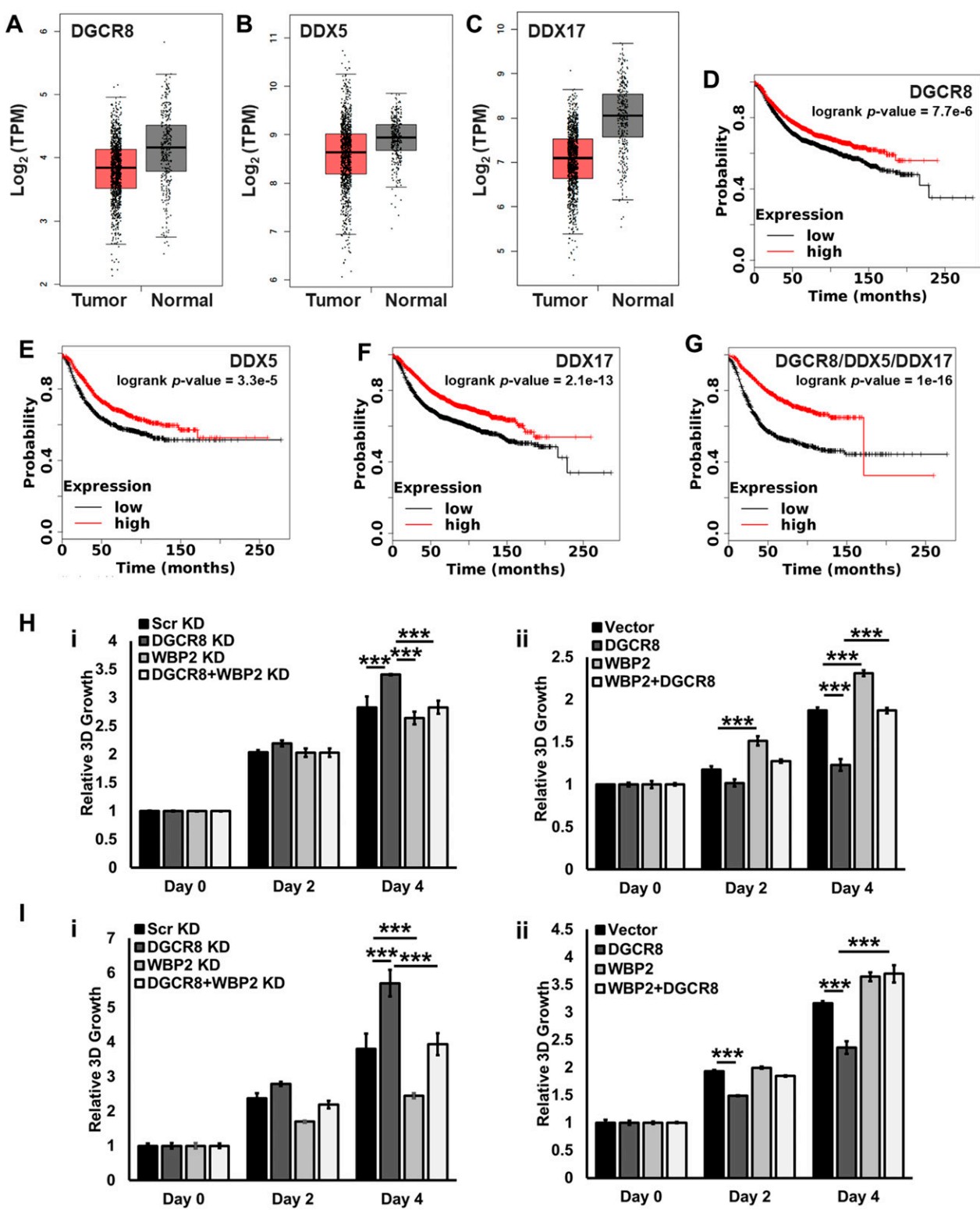

**Figure 6. The tumor-suppressive role of the microprocessor complex is inversely affected by WBP2.**
**(A, B, C)** Down-regulation of (A) DGCR8, (B) DDX5, and (C) DDX17 transcript levels in 1,085 breast tumors as compared with 291 controls (P-values < 0.01). **(D, E, F, G)** Analysis of the correlation of (D) DGCR8, (E) DDX5, (F) DDX17 and (G) *DGCR8/DDX5/DDX17* gene expression signature with the relapse-free survival of breast cancer samples. **(H)** Analysis of the 3D proliferation rate of MCF-7 cells upon DGCR8 knockdown or DGCR8/WBP2 double knockdown (i), or DGCR8 overexpression or DGCR8/WBP2 co-overexpression. **(I)** Analysis of the 3D proliferation rate of T47D cells upon DGCR8 knockdown or DGCR8/WBP2 double knockdown (i), or DGCR8 overexpression or DGCR8/WBP2 co-overexpression.

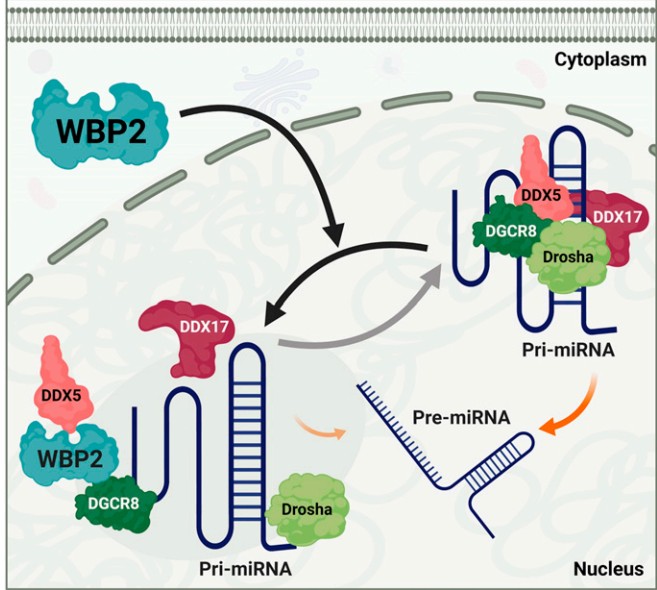

**Figure 7. The schematic model illustrating the regulation of pre-miRNA processing by WBP2 protein.**
Upon nuclear translocation, WBP2 interacts with the microprocessor complex that results in the disruption and subsequent inhibition of pri-miRNA processing in breast cancer cells.

microprocessor complex in breast cancer. However, DGCR8 protein has specific functions that are independent of Drosha. Hence, the tumor-suppressive role of DGCR8 might also not be necessary as a result of its miRNA-processing role. In fact, hundreds of mRNAs, small nucleolar RNAs (snoRNAs), and long noncoding RNAs (lncRNAs) are among the list of molecules, that undergo processing steps by DGCR8 protein independent of Drosha (32). Owing to the diverse range of uncommon functions of DGCR8 and Drosha in the cell, targeting the whole microprocessor complex via silencing all the components could better clarify the overall oncogenic or tumor-suppressive nature of this complex in the context of breast cancer.

The in vitro proliferation assays further showed that WBP2 could revert the proposed tumor-suppressive property of DGCR8. Although this finding supports the notion that WBP2 mediates this effect via inhibiting the miRNA-processing function of DGCR8, it is also possible that WBP2 regulated the DGCR8 growth-related effects via its involvement in other known signaling pathways, such as EGFR and $PI_3K/Akt$. Thus, the impact of WBP2 on DGCR8-mediated growth could be mediated by the diverse roles of this oncoprotein, which likely include its miRNA-processing inhibitory action, at least in part.

The identification of the key interacting domains of WBP2 and DGCR8 could be feasibly implemented by performing cross-linking mass spectrometry to narrow down to the broad patches/domains required for the interaction. Of note, subsequent site-directed mutational investigations in those residues could reveal whether the mutant DGCR8 could still affect the cell growth or how the mutant WBP2 could affect the DGCR8-mediated changes in cell proliferation. The biological

significance of the outcomes will shed light on how WBP2 manipulates cell proliferation via inhibiting the microprocessor complex activity.

Thus far, different auxiliary regulators of microprocessor complex activity have been reported, such as BRCA1 (23), Smads (24), Myc (25), YAP (27), ERα (33), and p53 (26). Of note, two well-studied WBP2-interacting proteins—YAP and ERα—like WBP2, negatively regulated the activity of the microprocessor complex. This suggests a testable hypothesis that WBP2 suppresses the microprocessor complex activity cooperatively with YAP and ERα. However, this is beyond the scope of this study.

In conclusion, WBP2 interacts with the components of the microprocessor complex, leading to the suppression of microprocessor complex assembly and its putative tumor suppressor function. Although we showed that the WBP2/microprocessor complex interaction(s)—and a subsequent microprocessor complex disassembly—is not mediated by WBP2's PPxY domain, the exact mechanism has remained to be fully mapped. The disassembling mechanism could be mediated by the direct interaction of WBP2 with at least one of the microprocessor complex components, such as DGCR8, or interaction between WBP2 with a third protein, which is crucial for the docking of microprocessor complex components, could restrict their bindings to the complex. Alternatively, the interactions between WBP2 and YAP or ERα that are known WBP2 partners with inhibitory effects on the microprocessor complex activity could be another potential mechanism. These hypotheses remained to be tested in this study.

The study offers new insights into the regulation of microprocessor complex activity by yet another novel auxiliary regulator—the WBP2 oncoprotein. This improves our understanding of the molecular etiology of breast cancer that may have implications in breast cancer therapy upon deeper molecular and clinical investigations.

## Materials and Methods

### Antibodies and reagents

The list of antibodies, siRNAs, and primers is detailed in the Supplemental Data 1 (Tables S1–S4).

### Cell culture

Human breast cancer epithelial cell lines T47D and MCF-7 were purchased from American Type Culture Collection. These cells were cultured in RPMI1640 containing 10% FBS and 100 U penicillin/ streptomycin. RPMI1640, DMEM, and FBS were purchased from Thermo Fisher Scientific Hyclone.

### Cell lysis and immunoblotting

Cells were washed twice with ice-cold PBS and lysed with non-ionic denaturing lysis buffer (50 mM Tris, pH7.5, 1 mM EDTA, 150 mM NaCl, 0.5% Triton X-100, 0.5% Nonidet-P40, and 10% glycerol), containing

protease and phosphatase inhibitor cocktail (Pierce). The cell lysates were vortexed vigorously and centrifuged at 16,000*g* for 15 min at 4°C. The protein concentration of the lysates was estimated using Bradford Ultra Protein Assay reagent (Expedeon). 40 *μg* proteins were resolved by SDS–PAGE on 10% gels. The resolved proteins were subsequently transferred onto polyvinyl difluoride (PVDF) membranes (Pierce). The membrane was blocked using 1% BSA in Tris-buffered saline in 1% Tween-20 (TBST). The blots were then incubated with antibodies at their optimal dilutions overnight at 4°C. The anti-rabbit or mouse HRP-conjugated secondary antibodies (Pierce) were diluted at 1:5,000 in blocking buffer. The blots were incubated with the secondary antibodies for 1 h at room temperature. Visualization was performed using the Pierce ECL Western Blotting substrate for whole cell lysate immunoblotting or Advansta Western Bright ECL HRP substrate for subcellular fractionations and post-immunoprecipitation immunoblotting.

## Subcellular fractionation

Nuclear and cytoplasmic extracts were prepared using the Nuclear Complex Co-IP Kit (Active Motif), as per the manufacturer's instructions.

## Co-immunoprecipitation (co-IP) assay

For the co-immunoprecipitation assay, the cytoplasmic and nuclear extracts were used. 300–1,000 *μg* cells were washed with ice-cold 1× PBS and lysed in nonionic denaturing lysis buffer. 1 mg of protein lysates were incubated with 2 *μg* of WBP2 antibody overnight at 4°C on a rotator. Immune complexes were then precipitated with 50 *μ*l of Dynabeads Protein G (Invitrogen) and incubated for 30 min at room temperature with rotation. Immunoprecipitates were washed twice (5 min per wash) with washing buffer I (50 mM Tris, pH 7.4, 1 mM EDTA, 500 mM NaCl, 0.5% Nonidet P-40, and 0.5% Triton X-100) followed by washing buffer II (50 mM Tris, pH 7.4, 150 mM NaCl, 0.5% Nonidet P-40, 1 mM EDTA, and 0.5% Triton X-100) at 4°C. Proteins were then eluted by boiling the beads in 30 *μ*l of 2× sample buffer with *β*-mercaptoethanol at 95°C for 10 min. The eluted proteins were subjected to immunoblotting.

## Generation of WBP2 localization mutant plasmid

GFP-WBP2 and GFP-C1 were kind gifts from Professor Hong Wanjin (A*STAR). Plasmids for pmTurquoise2-NES (#36206) and NLS-GFP (#67652) were obtained from Addgene. GFP-WBP2 (NES) and GFP-WBP2 (NLS) were constructed by cloning of mTurquoise2-NES (from pmTurquoise2-NES) and GFP-NLS (from NLS-GFP), respectively, to the C-terminal of GFP-WBP2. The localization control, GFP-WBP2 (WT), was generated by introducing a stop codon through point mutation and the N-terminal of NES of GFP-WBP2 (NES) via site-directed mutagenesis using the QuikChange Lightning multisite-directed mutagenesis kit (Stratagene; Agilent Technologies).

## Confocal microscopy and image analysis

Cells were transfected with the WBP2 localization mutant plasmids, and seeded on *μ*-Slide 8 Well (ibidi) and fixed with 4% para-formaldehyde (Electron Microscopy Sciences) diluted in PBS.

Samples were then imaged using the FV1000 confocal microscopy system (Olympus). Image analysis was performed using ImageJ. The average pixel intensity in the nucleus and cytoplasm were recorded. The localization of WBP2 (*L*) was calculated as followed:

$$L = Log_2\left(\frac{\overline{P}_N}{\overline{P}_C}\right),$$

where $\overline{P}_N$ is the average pixel intensity in the nucleus, and $\overline{P}_C$ is the average pixel intensity in the cytoplasm. As *L* is in a logarithmic scale, thus $L = 0$ when the $\overline{P}_N = \overline{P}_C$, while $L > 0$ and $L < 0$ corresponds to $\overline{P}_N > \overline{P}_C$ and $\overline{P}_N < \overline{P}_C$, respectively.

## Transient transfection

Cells were reverse-transfected with corresponding siRNAs (final concentration of 50 nM), miRNA mimic/inhibitor (final concentration of 20 nM), and plasmids (1–2 *μg*) using Jet PRIME reagent (Polyplus Transfection). Transfection was performed according to the manufacturer's instructions and cells were harvested at 24–48 h posttransfection.

## Dual-luciferase reporter assay

Reporter activity was measured using the Dual-Luciferase Reporter Assay (Promega) system according to the manufacturer's instruction. The quantification was carried out using Luminoskan Ascent Microplate Luminometer (Thermo Fisher Scientific). Renilla signals were normalized to Firefly.

## Microprocessor complex activity assay

The activity of the microprocessor complex was studied using psiCHECK2 incorporating pri-miR-125b-1 or pri-miR-205 downstream of the Renilla luciferase gene. Changes in the microprocessor complex activity were measured by the Dual-Luciferase Reporter Assay system. The ratio of Firefly/Renilla luminescence was used to quantify the activity of the microprocessor complex.

Alternatively, a qPCR-based methodology was carried out to assess the microprocessor complex activity using an independent approach. In this method, the activity of the microprocessor complex was measured quantitatively based on the production of given pre-miRNAs corresponding to its pri-miRNA level. Therefore, the pre-/pri-miRNA ratio reflects the activity of microprocessor complex activity (27). In this assay, the total RNA was first extracted. This was followed by the sub-population isolation of different RNA species, that is, >200-nucleotide and <200-nucleotide extracts (Fig S2A). *GAPDH* was used as the reference gene for quantification of RNAs with >200-nucleotide length, that is, pri-miRNA, whereas *U6 snRNA* was used as the control for RNAs with <200-nucleotide length, that is, pre-miRNA.

## In vitro cell-based assays

Cells were transiently transfected with conditions of interest in a six-well plate (day 0) and incubated overnight. On day 1, 1,500–2,000

cells/well were seeded in triplicate in a 96-well plate in 100 μl of culture medium. Once the cells adhered, cell proliferation was measured using the CellTiter 96 Aqueous One Solution Non-Radioactive (MTS) Cell Proliferation Assay Reagent (Promega). 20 μl of MTS reagent was added to each well and following incubation of 2 h the absorbance was measured at 490 nm using a plate reader. Cell proliferation was monitored for 5 d. For 3D proliferation assay, 10,000 cells/well were seeded in triplicate in an ultra-low-attachment 96-well plate and measured via CellTiter-Glo 3D Cell Viability Assay (Promega), according to the manufacturers' instructions.

### Correlation of gene expression levels in clinical specimens

Kaplan–Meier plotter (http://kmplot.com/analysis) was used to assess the effect of genes on survival using breast cancer patients' samples (34). GEPIA database (35) was used to study the quantitative expression profiles of the genes based on the analysis of TCGA and genotype-tissue expression (GTEx) projects RNA-seq data.

### Statistical analysis

Statistical package for the social sciences (SPSS) was used to perform all statistical analyses. $t$ test was used for comparisons between two groups. Bar graph represents quantification as mean ± SEM of three biological replicates. The indicated significance values correspond to <0.05 (∗), <0.01 (∗∗), and <0.001 (∗∗∗).

## Data Availability

Data sharing not applicable to this article as no datasets were generated or analyzed during the current study.

## Supplementary Information

## Acknowledgements

This work is supported by funding from the Ministry of Education, Singapore (MOE2016-T2-2007). The schematic figure (Fig 7) was created with BioRender.com under the National University of Singapore subscription.

### Author Contributions

H Tabatabaeian: conceptualization, formal analysis, investigation, visualization, methodology, and writing—original draft, review, and editing.
SK Lim: formal analysis, investigation, and methodology.
T Chu: investigation and methodology.
SH Seah: investigation and methodology.
YP Lim: conceptualization, supervision, and writing—review and editing.

### Conflict of Interest Statement

The authors declare that they have no conflict of interest.

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
