## [Reviewer comments · Life Science Alliance]

Life Science Alliance

WBP2 inhibits microRNA biogenesis via interaction with the microprocessor complex

Hossein Tabatabaeian, Shen Kiat Lim, Tinghine Chu, Sock Hong Seah, and Yoon Pin Lim
DOI: <https://doi.org/10.26508/lsa.202101038>

Corresponding author(s): Yoon Pin Lim, National University of Singapore

Review Timeline:

Submission Date:	2021-01-27
Editorial Decision:	2021-03-26
Revision Received:	2021-04-08
Editorial Decision:	2021-05-19
Revision Received:	2021-05-26
Accepted:	2021-05-28

Scientific Editor: Shachi Bhatt

Transaction Report:

March 26, 2021

Re: Life Science Alliance manuscript #LSA-2021-01038

Dr. Yoon Pin Lim
National University of Singapore
5 Science Drive 2, Blk MD4, Level 3, Singapore 117545
Singapore 117545
Singapore

Dear Dr. Lim,

Thank you for submitting your manuscript entitled "WBP2 inhibits microRNA biogenesis via interaction with the microprocessor complex" to Life Science Alliance. The manuscript was assessed by expert reviewers, whose comments are appended to this letter.

We apologize for this unusual and extended delay in getting back to you. As you will note from the reviewers' comments below, the reviewers are interested in these findings, but have also raised a number of significant questions that should be addressed prior to further consideration of the manuscript at LSA. We would, thus, encourage you to submit a revised version of the manuscript that addresses all of the reviewers' points.

Thank you for this interesting contribution to Life Science Alliance. We are looking forward to receiving your revised manuscript.

Sincerely,

Shachi Bhatt, Ph.D.

Executive Editor

Life Science Alliance

<https://www.lsjournal.org/>

Interested in an editorial career? EMBO Solutions is hiring a Scientific Editor to join the international Life Science Alliance team. Find out more here -

https://www.embo.org/documents/jobs/Vacancy_Notice_Scientific_editor_LSA.pdf

B. MANUSCRIPT ORGANIZATION AND FORMATTING:

Reviewer #1 (Comments to the Authors (Required)):

In this manuscript, Tabatabaeian and cols study the roles of the protein WWBP2 in the regulation of the assembly and activity of the microprocessor activity in cancer cell lines. The authors show that WBP2 limits the activity of the microprocessor complex affecting the processing of pri-miRNAs. Changes in the localization of this protein affect its activity supporting its role in the nucleus. The

authors also perform co-IP experiments and find that WBP2 physically interact with this complex and downregulates its activity.

Given that miRNAs are central regulators of gene activity in normal development, tissue homeostasis, as well as in cancer, we need to understand the molecular elements controlling miRNA activity. This study is therefore of general interest.

The manuscript is well-written, structured, and presented. In general, the conclusions obtained are well supported by the results. However, before the manuscript is ready for publication, the authors should address some concerns.

In Fig 6, the authors show how WBP2 manipulation affects the growth rate of cells over-expressing and depleting DGRC8. The authors should show in this set up the growth properties over-expressing and downregulating WBP2 in that setup. This would allow to compare with the double-overexpression and double-KD condition and provide a better insight on whether the effect observed is specific on that complex or is an additive effect due to independent growth regulatory roles of this protein.

In line with that comment, WBP has been shown to control other signaling pathways that have profound implications in cell proliferation and cancer. The results presented here do not prove that the growth-related effects observed upon WBP2 manipulation are due to miRNA activity or might be caused by a more general role in growth regulation. The authors should consider this as a possibility that, in my view, should be mentioned in the discussion.

Minor comments:

Page 7: "DGCR8, which is a key component of the microprocessor complex, was used as a positive control." I guess what the authors mean is that DGCR8-KD was used as a control. If that's the case, please, correct it.

Page 8: "miR-19a and miR-19b were previously determined to be incapable of targeting WBP2, while miR-23a had been proven to target WBP2, miR-205 was randomly selected." Where has this been shown? If it has been previously published, the authors should cite the corresponding source.

Reviewer #2 (Comments to the Authors (Required)):

In their manuscript "WBP2 inhibits microRNA biogenesis via interaction with the microprocessor complex" Tabatabaieian et al. have explored the importance of WBP2 protein in the miRNA biogenesis regulation in mammalian cancer and other cells to conclude a WBP2 mediated problem in microprocessor complex assembly to cause a retarded pri-miRNA processing in presence of WBP2. They have further showed the importance of WBP2-mediated regulation of miRNA biogenesis in cancer cell growth. I have the following comments to be considered by the authors to make this manuscript acceptable for publication in Life Science Alliance.

Major Comments:

1. The author used a luciferase activity measurement based reporter to assess the miRNA processing event to score the changes throughout the study. It would be good to have Northern Blot data to validate the results at least in couple of context defined in Figure 2E.
2. It would be good to know the effect that the WBP2 has on mature miRNA levels. Does WBP2 affect the mature and Ago-associated miRNA levels? This is important as majority of the WBP2

effect should be due to an ultimate reduction in miRNA formation in WBP2 over expression context that supposed to affect the microprocessor mediated pri- to pre-miRNA conversion.

3. It is not clear what molecular targets that are getting affected in WBP2 KD conditions to change cancer cell growth described in Figure 6. The proliferative or senescent status change could be measured to know what has happened with those cells. It would be essential to know the causative role of pri-miRNA processing defect on cell growth. Do these two processes are coupled?

4. It would be good to know the domain of WBP2 that by interacting with DGCR8 affect the microprocessor complex assembly. A mutant version of WBP2 with truncation of DGCR8 interacting domain can be used to show how the WBP2 mediated cancer cell growth is related to its interaction with DGCR8.

Minor comments:

It would be good to have more information in the legends of the figures to enable the readers to understand the paper easily.

Reviewer #3 (Comments to the Authors (Required)):

The manuscript by Tabatabaeian et al describes the role of WBP2 in inhibiting the formation of microprocessor complex thereby regulating the microRNA biogenesis. They show that the loss of WBP2 leads to enhanced processing of microRNA pre-cursor transcripts and that over-expression leads to abrogation of tumor-suppressive functions of microprocessor complex. The required experiments have been performed to support their claim but further confirmations are required to conclude the same.

Fig. 1 Ai and ii, western blot. The DGCR8 KD does not result in appreciable decrease of the DGCR8 protein level (i) but does in (ii), though the assay was done in same (MCF7) cells. This could simply be a different KD efficiency in different experiments, but what is concerning is that the decrease in Luciferase activity seems to be affected by similar levels in i and ii. This raises the question that if the decreased Luciferase activity is dependent on DGCR8 or is an artifact. Please clarify.

Fig. 2D, Could the authors also quantify specific mature microRNAs by taqman assays and show that not only the pri-miRNA transcripts decrease but also the mature miRNAs.

Fig. 4. The legend is wrong and refers to wrong panels. Please correct it. Reg, 4Bii and 4C, D, I have a concern with regard to stoichiometry of the proteins when over expressed. Would forced over expression of any given two proteins lead to their interaction? Here, I would like to see some unrelated protein over expressed and that there is no interaction with the unrelated protein and microprocessor complex. Ideally, the authors could choose a cell line where WBP2 is sufficiently expressed to be able to pull down and look for the microprocessor complex either by western blot or even my Mass Spectrometry which could be more sensitive.

Page 12, second line is incomplete. I believe the authors wanted to write "similarly reduced the luciferase activity"?

Supp. Fig. 3. It would be nice to know the interface or the domains involved in the interaction between WBP2 and the components of the microprocessor complex and then by inhibiting the interaction specifically, the biological significance of such interaction could be estimated. The authors have tried to abrogate the interaction using mutations in PY motif, which unfortunately did

not yield the desired results. Here, I would like to suggest that crosslinking Mass-spectrometry could potentially be used to narrow down to the broad patches/domains required for the interaction.

Discussion, 4th line, "we recently showed"

Last sentence, please tone down, it goes far beyond what this manuscript shows

Adding line numbers would be easy for reviewing.

Reviewer #1

In this manuscript, Tabatabaeian and cols study the roles of the protein WBP2 in the regulation of the assembly and activity of the microprocessor activity in cancer cell lines. The authors show that WBP2 limits the activity of the microprocessor complex affecting the processing of pri-miRNAs. Changes in the localization of this protein affect its activity supporting its role in the nucleus. The authors also perform co-IP experiments and find that WBP2 physically interact with this complex and downregulates its activity.

Given that miRNAs are central regulators of gene activity in normal development, tissue homeostasis, as well as in cancer, we need to understand the molecular elements controlling miRNA activity. This study is therefore of general interest.

The manuscript is well-written, structured, and presented. In general, the conclusions obtained are well supported by the results. However, before the manuscript is ready for publication, the authors should address some concerns.

1- In Fig 6, the authors show how WBP2 manipulation affects the growth rate of cells over-expressing and depleting DGCR8. The authors should show in this set up the growth properties over-expressing and downregulating WBP2 in that setup. This would allow to compare with the double-overexpression and double-KD condition and provide a better insight on whether the effect observed is specific on that complex or is an additive effect due to independent growth regulatory roles of this protein.

Response 1: Thank you for this important comment. We added the data on the effect of WBP2 knockdown and overexpression on 3D proliferation growth of MCF-7 and T47D cells (Fig. 6H & I). Briefly, DGCR8 KD resulted in increased 3D growth, while WBP2 KD decreased, specifically in T47D cells. DGCR8/WBP2 double-KD abolished the DGCR8 depletion-mediated cell growth. The consistent results were obtained from the overexpression panel, where DGCR8 overexpression decreased the 3D growth – opposite of WBP2 overexpression. Co-overexpression of WBP2 and DGCR8 rescued the inhibitory effect of DGCR8 overexpression. These data suggest that WBP2 could negatively affect the DGCR8 tumor-suppressive function. The text has been modified accordingly on page 12, lines 199-210.

2- In line with that comment, WBP has been shown to control other signaling pathways that have profound implications in cell proliferation and cancer. The results presented here do not prove that the growth-related effects observed upon WBP2 manipulation are due to miRNA activity or might be caused by a more general role in growth regulation. The authors should consider this as a possibility that, in my view, should be mentioned in the discussion.

Response 2: Thank you for this comment. We have taken this possibility into account by adding a paragraph into the discussion section (page 15, lines 279 to 285).

3- Page 7: "DGCR8, which is a key component of the microprocessor complex, was used as a positive control." I guess what the authors mean is that DGCR8-KD was used as a control. If that's the case, please, correct it.

Response 3: Thank you for the comment. As you mentioned, we knocked DGCR8 down to use it as a positive control to show the credibility of the assay we performed to assess the microprocessor complex activity. The sentence has been revised for better clarity (page 6, line 64).

4- Page 8: "miR-19a and miR-19b were previously determined to be incapable of targeting WBP2, while miR-23a had been proven to target WBP2, miR-205 was randomly selected." Where has this been shown? If it has been previously published, the authors should cite the corresponding source.

Response 4: Thank you for the comment. The related reference has been added accordingly. Please refer to page 7, line 96

Reviewer #2

In their manuscript "WBP2 inhibits microRNA biogenesis via interaction with the microprocessor complex" Tabatabaeian et al. have explored the importance of WBP2 protein in the miRNA biogenesis regulation in mammalian cancer and other cells to conclude a WBP2-mediated problem in microprocessor complex assembly to cause a retarded pri-miRNA processing in presence of WBP2. They have further showed the importance of WBP2-mediated regulation of miRNA biogenesis in cancer cell growth. I have the following comments to be considered by the authors to make this manuscript acceptable for publication in Life Science Alliance.

1. The author used a luciferase activity measurement-based reporter to assess the miRNA processing event to score the changes throughout the study. It would be good to have Northern Blot data to validate the results at least in couple of context defined in Figure 2E.

Response 1: Thank you for the comment you raised. We do agree that more assays would have been better to validate the luciferase activity measurement-based reporter to assess the miRNA processing event. That is why we had used another methodology based on qPCR technique to re-assess the authenticity of the results. In this method, we designed primers for pre-miRNA/pri-miRNA and extracted pri-miRNA and pre-miRNA-containing RNA subpopulations and performed qPCR assay to measure the pre-miRNA/pri-miRNA ratio as a standard representative of microprocessor complex activity. The methodology was explained in detail in Materials and Methods file. As shown in **Fig. 2D**, the data were consistent with luciferase assay outcomes showing that WBP2 negatively regulates the pri-miRNA processing. As there was no contradiction between the two tests, we did not perform Northern blot. However, we have added this suggestion in the discussion section to consider this important comment (Page 13, lines 225-228).

2. It would be good to know the effect that the WBP2 has on mature miRNA levels. Does WBP2 affect the mature and Ago-associated miRNA levels? This is important as majority of the WBP2 effect should be due to an ultimate reduction in miRNA formation in the WBP2 overexpression context that is supposed to affect the microprocessor mediated pri- to pre-miRNA conversion.

Response 2: Thank you for the excellent comment. We used the same RNA samples obtained from the WBP2 KD/OE samples shown in **Fig. 2D**. Probing for the mature miRNAs, the qPCR results consistently showed that WBP2 down-regulated the selected miRNAs upon overexpression. Consistently, WBP2 depletion elevated the expression level of mature miRNAs. These data strongly supported our findings that WBP2 negatively regulates the miRNA biogenesis; thus, we have updated **Fig. 2** and the new results are presented as **Fig. 2E**.

3. It is not clear what molecular targets that are getting affected in WBP2 KD conditions to change cancer cell growth described in Figure 6. The proliferative or senescent status change could be measured to know what has happened with those cells. It would be essential to know

the causative role of pri-miRNA processing defect on cell growth. Do these two processes are coupled?

Response 3: We appreciate the concern raised by the reviewer. Since WBP2 has involvement in a wide range of cell proliferation-related signaling pathways, e.g. EGFR, Wnt, PI₃K/Akt and Hippo, the observations demonstrating that WBP2 reverted the effect of DGCR8 on cell proliferation do not directly prove that such effect is mediated by inhibiting the miRNA processing. Thus, we explicated the possibilities to clarify the effect of WBP2 on the DGCR8-related cell growth is likely to be due to a myriad of reasons including its effect on miRNA processing, at least in part. This can be found on page 15, lines 279-285. On the other hand, and as discussed in page 14, paragraph 3 to page 15, it may be not meaningful to dissect the role of pri-miRNA processing in cancer growth. This is because due to the diverse range of functions of DGCR8 and Drosha, the overall oncogenic or tumor-suppressive nature of microprocessor complex in the context of breast cancer is likely to be a result of the summation of all the key components being targeted, positively and negatively.

4. It would be good to know the domain of WBP2 that by interacting with DGCR8 affect the microprocessor complex assembly. A mutant version of WBP2 with truncation of DGCR8 interacting domain can be used to show how the WBP2 mediated cancer cell growth is related to its interaction with DGCR8.

Response 4: Thank you for this suggestion. Unfortunately, our data only showed that the PPxY motifs of WBP2, which are responsible for WBP2's interactions, did not play any role to interact with DGCR8. The PPxY mutants also did not affect the microprocessor complex activity significantly, as compared to the wild-type WBP2 (Supp. Fig.4). Further in-depth studies on WBP2 and DGCR8 domains represent a substantial amount of work that we feel is more suitable for a separate study to map the underlying domains responsible for the interaction between WBP2 and DGCR8 and subsequent functional assays. However, we have included this good suggestion in the discussion section. This can be found on page 15, line 286 to page 16, line 292.

5- It would be good to have more information in the legends of the figures to enable the readers to understand the paper easily.

Response 5: Thank you for the comment. We have improved the legends by adding more details.

Reviewer #3

The manuscript by Tabatabaeian et al describes the role of WBP2 in inhibiting the formation of microprocessor complex thereby regulating the microRNA biogenesis. They show that the loss of WBP2 leads to enhanced processing of microRNA pre-cursor transcripts and that over-expression leads to abrogation of tumor-suppressive functions of microprocessor complex. The required experiments have been performed to support their claim but further confirmations are required to conclude the same.

1- Fig. 1 Ai and ii, western blot. The DGCR8 KD does not result in appreciable decrease of the DGCR8 protein level (i) but does in (ii), though the assay was done in same (MCF7) cells. This could simply be a different KD efficiency in different experiments, but what is concerning is that the decrease in Luciferase activity seems to be affected by similar levels in i and ii. This raises the question that if the decreased Luciferase activity is dependent on DGCR8 or is an artifact. Please clarify.

Response 1: Thank you for the comment. As you mentioned, the reason for the different silencing levels could be due to the efficiency of siRNA transfection in different experiments. Using the same siRNA to repeat the microprocessor complex assay gave us around a 40-60% decrease in protein expression although the extent of the pri-miRNA processing seemed to be more consistently affected. Moreover, we used DGCR8 overexpression as the positive control in the subsequent experiments that consistently showed the credibility of the assay we performed. Considering all the results obtained from DGCR8 knockdown and overexpression, we concluded that the construct we designed and used to assess the microprocessor complex activity worked robustly, which responded to the KD and OE of DGCR8 in an expected manner. We have added the explanation to the text to explain why DGCR8 KD was replaced by OE as the positive control of microprocessor complex activity assay (page 6, lines 64-67).

2- Fig. 2D, Could the authors also quantify specific mature microRNAs by taqman assays and show that not only the pri-miRNA transcripts decrease but also the mature miRNAs.

Response 2: Thank you for the excellent comment. We used the same RNA samples obtained from the WBP2 KD/OE samples shown in **Fig. 2D**. Probing for the mature miRNAs, the qPCR results consistently showed that WBP2 down-regulated the selected miRNAs upon overexpression. Consistently, WBP2 depletion elevated the expression level of mature miRNAs. These data strongly supported our findings showing that WBP2 negatively regulates the miRNA biogenesis; thus, we have updated **Fig. 2** and the new results are presented as **Fig. 2E**.

3- Fig. 4. The legend is wrong and refers to wrong panels. Please correct it. Reg, 4Bii and 4C, D, I have a concern with regard to stoichiometry of the proteins when over expressed. Would forced over expression of any given two proteins lead to their interaction? Here, I would like to see

some un-related protein over expressed and that there is no interaction with the unrelated protein and microprocessor complex. Ideally, the authors could choose a cell line where WBP2 is sufficiently expressed to be able to pull down and look for the microprocessor complex either by western blot or even my Mass Spectrometry which could be more sensitive.

Response 3: Thank you for the comment. The legend has been revised accordingly.

As for the pull-down assays, the nuclear expression of WBP2 in the cell lines with higher endogenous expression of this protein was still too low to perform immunoprecipitation analyses. This is because the nuclear population of WBP2 represents only a very small fraction of total WBP2.

Regarding the possibility that forced overexpression of any given two proteins could lead to their interaction, our current data in Fig. 4Bii show that DDX17 did not co-precipitate with WBP2 in MCF-7 even though its expression in nuclear is quite high. Moreover, Drosha or DDX17 pull-down did not show any interaction with WBP2 in Fig. 4D. Considering these observations as the internal controls, the current data suggest that the interactions observed are specific. Besides, since the pull-down of both WBP2 and the reciprocal microprocessor complex components resulted in similar results, and given the effect of WBP2 overexpression/knockdown on the microprocessor complex assembly (**Fig. 5**), we concluded that the co-IP results were convincing. We, however, have discussed using mass spectrometry to map the WBP2's interactions with the microprocessor complex at the endogenous level to further support the claims (Page 13, line 237 to page 14, line 241).

4- Page 12, second line is incomplete. I believe the authors wanted to write "similarly reduced the luciferase activity"?

Response 4: Thank you for your attention. The sentence has been rectified.

5- Supp. Fig. 3. It would be nice to know the interface or the domains involved in the interaction between WBP2 and the components of the microprocessor complex and then by inhibiting the interaction specifically, the biological significance of such interaction could be estimated. The authors have tried to abrogate the interaction using mutations in PY motif, which unfortunately did not yield the desired results. Here, I would like to suggest that crosslinking Mass-spectrometry could potentially be used to narrow down to the broad patches/domains required for the interaction.

Response 5: Thank you for this valuable comment. Due to technical limitations, unfortunately, we are unable to perform the suggested work. However, the authors agree that the proposed work could pave the way towards a better understanding of the molecular interaction between the WBP2 and the microprocessor complex. Thus, we have a discussion on this on page 15, line 286 to page 16, line 292.

6- Discussion, 4th line, "we recently showed"

Response 6: The grammatical error has been fixed.

7- Last sentence, please tone down, it goes far beyond what this manuscript shows.

Response 7: Thank you for this comment. The last sentence has been revised accordingly.

8- Adding line numbers would be easy for reviewing.

Response 8: The line numbers have been added to the main text body accordingly.

May 19, 2021

RE: Life Science Alliance Manuscript #LSA-2021-01038R

Dr. Yoon Pin Lim
National University of Singapore
5 Science Drive 2, Blk MD4, Level 3, Singapore 117545
Singapore 117545

Dear Dr. Lim,

Thank you for submitting your revised manuscript entitled "WBP2 inhibits microRNA biogenesis via interaction with the microprocessor complex". We would be happy to publish your paper in Life Science Alliance pending final revisions necessary to meet our formatting guidelines.

Along with the points listed below, please also attend to the following:

- please add ORCID ID for the corresponding author-you should have received instructions on how to do so
- we encourage you to revise the figure legend for figure S1 as there is only one panel and it is not necessary to be labeled (you may remove it from the actual figure, as well)
- we encourage you to revise the figure legends for figures 4 and 5 such that the figure panels are introduced in alphabetical order (D is missing)
- please add your table legends to the main manuscript text after the figure legends
- there are callouts for figure 5A-D although the panels are not introduced in the actual figure
- please add callouts for Figures 5Ai, Bi, Ci, Di; S4A-C; S6A-D to your main manuscript text
- please provide higher resolution, higher quality images for blots shown in Figure S4B

A. FINAL FILES:

-- High-resolution figure, supplementary figure and video files uploaded as individual files: See our detailed guidelines for preparing your production-ready images, <https://www.life-science->

alliance.org/authors

B. MANUSCRIPT ORGANIZATION AND FORMATTING:

Sincerely,

Shachi Bhatt, Ph.D.
Executive Editor
Life Science Alliance
<http://www.lsjournal.org>
Tweet @SciBhatt @LSAJournal

Reviewer #1 (Comments to the Authors (Required)):

The authors have addressed all my concerns and the paper is now ready for publication.

Reviewer #2 (Comments to the Authors (Required)):

The revised manuscript have now addressed my major criticism and now acceptable for publication

Reviewer #3 (Comments to the Authors (Required)):

First of all, Apologies for the delay. The authors have mostly addressed this reviewers questions and could not perform one of the suggested experiment, which I think is ok, because of technical difficulties. I have no further questions and in my opinion, the manuscript can now be accepted.

May 28, 2021

RE: Life Science Alliance Manuscript #LSA-2021-01038RR

Dr. Yoon Pin Lim
National University of Singapore
5 Science Drive 2, Blk MD4, Level 3, Singapore 117545
Singapore 117545
Singapore

Dear Dr. Lim,

Thank you for submitting your Research Article entitled "WBP2 inhibits microRNA biogenesis via interaction with the microprocessor complex". It is a pleasure to let you know that your manuscript is now accepted for publication in Life Science Alliance. Congratulations on this interesting work.

DISTRIBUTION OF MATERIALS:

Again, congratulations on a very nice paper. I hope you found the review process to be constructive and are pleased with how the manuscript was handled editorially. We look forward to future exciting submissions from your lab.

Sincerely,

Shachi Bhatt, Ph.D.
Executive Editor
Life Science Alliance
<http://www.lsjournal.org>
Tweet @SciBhatt @LSAJournal